# Circular RNA Expression Profiles in Plasma from Patients with Heart Failure Related to Platelet Activity

**DOI:** 10.3390/biom10020187

**Published:** 2020-01-25

**Authors:** Yeying Sun, Xiaoli Jiang, Yan Lv, Xinyue Liang, Bingrui Zhao, Weihua Bian, Daolai Zhang, Jing Jiang, Chunxiang Zhang

**Affiliations:** 1US-China Institute for Translational Medicine, College of Pharmacy, Binzhou Medical University, Yantai 264000, China; sunyy@bzmc.edu.cn (Y.S.); 18363889625@163.com (X.J.); sunyy21cn@163.com (X.L.); sunyy21cn@aliyun.com (B.Z.); bian_1005@163.com (W.B.); dlzhang313@163.com (D.Z.); jing_jiang1974@sina.com (J.J.); 2Department of Cell Biology, College of Life Science, Yantai University, Yantai 264005, China; huayuanly@163.com

**Keywords:** circRNAs, platelet activity, heart failure

## Abstract

Heart failure (HF) is a deadly disease that is difficult to accurately diagnose. Circular RNAs (circRNAs) are a novel class of noncoding RNAs that might play important roles in many cardiovascular diseases. However, their role in HF remains unclear. CircRNA microarrays were performed on plasma samples obtained from three patients with HF and three healthy controls. The profiling results were validated by quantitative reverse transcription polymerase chain reaction. The diagnostic value of circRNAs for HF was evaluated by receiver operating characteristic (ROC) curves. The expression profiles indicated that 477 circRNAs were upregulated and 219 were downregulated in the plasma of patients with HF compared with healthy controls. Among the dysregulated circRNAs, hsa_circ_0112085 (*p* = 0.0032), hsa_circ_0062960 (*p* = 0.0006), hsa_circ_0053919 (*p* = 0.0074) and hsa_circ_0014010 (*p* = 0.025) showed significantly higher expression in patients with HF compared with healthy controls. The area under the ROC curve for hsa_circ_0062960 for HF diagnosis was 0.838 (*p* < 0.0001). Correlation analysis showed that the expression of hsa_circ_0062960 was highly correlated with B-type natriuretic peptide (BNP) serum levels. Some differential circRNAs were found to be related to platelet activity by Gene Ontology (GO) and Kyoto Encyclopedia of Genes and Genomes (KEGG) pathway analyses. The landscape of circRNA expression profiles may play a role in HF pathogenesis and improve our understanding of platelet function in HF. Moreover, hsa_circ_0062960 has potential as a novel diagnostic biomarker for HF.

## 1. Introduction

Heart failure (HF), often known as congestive HF, occurs when the heart fails to properly pump sufficient blood to the whole body. HF is a major public health concern and is a leading cause of morbidity and mortality worldwide, affecting over 23 million worldwide and resulting in 300,000 annual deaths [1,2]. Accurate and timely diagnosis of HF is essential to ensure appropriate treatment for patients. However, diagnosing HF can be difficult when conventional tools are used [3]. The symptoms and signs of HF commonly overlap with those of other diseases and may be a few characteristics early in the disease process [4]. B-Type natriuretic peptide (BNP) or N-terminal pro BNP (NT-proBNP) is a widely used biomarker in HF and is primarily used to help detect, diagnose, and evaluate the severity of HF [5,6,7]. Given that BNP is involved in several physiological processes, the expression levels of plasma BNP or NT-proBNP are affected by aging, gender, and some cardiac function and endocrine diseases, such as hyperthyroidism [8,9]. BNP levels may also be altered by some drugs, such as angiotensin-converting enzyme inhibitors and diuretics, which increase fluid loss [9]. These potential factors may influence the accurate diagnosis of HF. Therefore, novel biomarkers are needed in HF diagnosis.

As a novel kind of noncoding RNAs, circular RNAs (circRNAs) have structures that differ from those of linear RNA [10,11]. Covalently closed-loop structures are formed by back-spliced circularization without 5’ and 3’ ends [12,13]. CircRNAs are more stable than linear RNAs because the former are resistant to RNase R [14,15]. CircRNAs are involved in important biological and development processes and show strong biological functions. Many circRNAs show strong regulation when gene expression is compared between normal and failing heart tissues [14,16]. Furthermore, circRNAs are circulated in large amounts in the bloodstream [17]. Given these unique features, circRNAs show great potential as biomarkers of cardiac disease. For example, the expression level of myocardial infarction-associated circular RNA (MICRA) in blood cells can predict left ventricular remodeling after acute myocardial infarction (AMI) [18]. In a large group, MICRA was proven to have good predictive value. [18]. The circRNAs can potentially aid in heart disease prediction.

In this research, we identified differentially expressed circRNAs in plasma of patients with HF. Some of these circRNAs may serve important functions in the platelet activation of patients with HF. Hsa_circ_0062960 might serve as a potential biomarker for HF diagnosis.

## 2. Materials and Methods 

### 2.1. Clinical Population and Control Groups

Thirty patients with chronic stable HF were recruited from the cardiovascular department of Yantai Affiliated Hospital of Binzhou Medical University from September 2016 to March 2017. Subjects who were age, gender, and body mass index matched to the HF group were recruited from the physical examination department and served as the control group. Four stages of HF were identified according to the guidelines of New York Heart Association (NYHA) functional classification by cardiologists specialized in HF. The control group needed to show an absence of a known cardiovascular disease with a standard of BNP < 100 (pg/mL), EF > 50%, and no clinical features. The clinical features of the patients and control groups are listed in Table 1. This study was approved by the institutional review committee of the Binzhou Medical University (approval number 2016-56), and the participants gave informed consent, thereby conforming to the Declaration of Helsinki.

### 2.2. Plasma Isolation and RNA Extraction

HF and control subjects were phlebotomized. Blood samples of HF patients were obtained three days before and after the antiplatelet drug aspirin treatment. A peripheral blood sample (10 mL) was collected into blood collection tubes (BD Vacutainer, Franklin Lakes, NJ, USA) that contained 3.8% sodium citrate. The blood samples were centrifuged at 2000× *g* for 10 min at 22 °C to obtain plasma. Plasma was stored at –80 °C in a new tube. Plasma (200 µL) was incubated at 37 °C in a water bath until the samples were completely thawed and salts were dissolved. To extract the circulating RNAs from the plasma samples, we used the miRNeasy Serum/Plasma Kit (Qiagen, Hilden, Germany) in accordance with the manufacturer’s instructions. About 200 μL of plasma was used for each RNA preparation sample. Total RNA on the spin column membrane was eluted by 14 μl of RNase-free water.

### 2.3. CircRNA Microarray Expression Profiling

The total RNA concentration and purity were evaluated by a NanoDrop One spectrophotometer (NanoDrop Technologies, Wilmington, DE, USA). RNA integrity was assayed with Bioanalyzer 2100 (Agilent Technologies, Palo Alto, CA, USA). According to the manufacturer’s description, the extracted RNAs were labeled with Cy3-dCTP after digestion, dephosphorylation, denaturation, and amplification. Agilent human circRNA Array (V2.0) ( Agilent, Santa Clara, CA, USA) containing 170,340 human circRNA probes was hybridized with the purified RNAs. We analyzed the microarray data using GeneSpring software V13.0 (Agilent Technologies, Santa Clara, CA, USA) to identify the differential circRNAs. Threshold values of ≥2- and ≤2-fold change and a *t*-test *p*-value of 0.05 were used. All the identified circRNAs were listed in the deposited online data [19].

### 2.4. CircRNA Expression Analysis

Quantitative reverse transcription polymerase chain reaction (qRT-PCR) was performed to verify the differential expression levels in the HF and the control groups using specific divergent primers designed for selected circRNAs. Primer Premier 5.0 software was used to design divergent primers based on microarray data. Total RNAs from the HF and control groups were reverse-transcribed to cDNA using the PrimeScript RT reagent kit (Takara, Dalian, China). According to the manufacturer’s instructions, qRT-PCR was performed using SYBR Premix Ex Taq™ II (Takara, Dalian, China). U6 small nuclear RNA (snRNA) was used as an internal control. Fluorescent signals were collected and normalized to an internal reference, and the threshold cycle was set within the exponential phase of PCR. Relative gene expression was calculated by comparing the cycle times for each target PCR, and each sample was tested in triplicate. The sequence and product size of PCR primers are listed in Table 2. To eliminate false-positive results, the product size of divergent primers was validated by agarose gel electrophoresis.

### 2.5. Gene Ontology and Kyoto Encyclopedia of Genes and Genomes Pathway Analyses

Gene Ontology (GO) enrichment analysis was performed by the GOseq R packages based on the circRNA-host genes. The Kyoto Encyclopedia of Genes and Genomes (KEGG) pathways was used to test the statistical enrichment of the circRNA-host genes by using KOBAS 3.0 software [20]. The lower the *p*-value was, the more significant the pathway and GO term were. A *p*-value of ≤0.05 was recommended.

### 2.6. Statistical Analysis

All data were presented as the mean ± standard error. For the relative gene expression of circRNAs, the horizontal lines represent the medians. The clinical diagnostic value of a special circRNA was evaluated by receiving operator characteristic (ROC) curve analysis in which an area under the curve (AUC) = 0.5 means no diagnostic value. ROC curve analysis also helped identify the cutoff value, and corresponding sensitivity and specificity could also be identified. Pearson’s correlation test was used to evaluate the correlation between the circRNA expression level and HF. Two-tailed unpaired Student’s *t*-tests and ANOVA were used for the statistical evaluation of the data. SPSS 19.0 (IBM Corporation, Armonk, NY, USA)was used for data analysis. *p* < 0.05 was considered significant.

## 3. Results

### 3.1. Expression Profiles of CircRNAs in the Plasma from Patients with HF

To investigate the circRNA expression profiles in healthy individuals and patients with HF, we performed microarray analysis of the expression profiles of circRNAs in plasma using the Agilent human circRNA Array V2.0. Hierarchical clustering and volcano plots were used to show the variations in circRNA expression between the HF and control groups (Figure 1). The expression levels of circulating circRNAs showed a significant difference between the HF and control groups. A total of 696 circRNAs were differentially expressed amongst them, 477 were upregulated, and 219 were downregulated in patients with HF. To obtain the most clinically applicable biomarker, we selected candidate biomarkers from the 477 upregulated circRNAs by utilizing the following screening criteria: fold changes ≥ 2.3, *p* ≤ 0.01, and raw intensity values ≥ 300 for each sample. Five circRNAs, namely hsa_circ_0112085, hsa_circ_0062960, hsa_circ_0011464, hsa_circ_0053919, and hsa_circ_0014010, met the standards and were selected for further analysis.

### 3.2. Verification of the CircRNA Profile by qRT-PCR

To validate the five selected candidate circRNAs, we performed qRT-PCR in a cohort consisting of 30 control subjects (*n* = 30) and 30 patients with HF. The expression levels of hsa_circ_0112085 (*p* = 0.0032), hsa_circ_0062960 (*p* = 0.0006), hsa_circ_0053919 (*p* = 0.0074), and hsa_circ_0014010 (*p* = 0.025) were significantly higher in the HF group than in the controls, which showed the same result with the microarray data (Figure 2). No significant difference in the expression levels of hsa_circ_0011464 was found between the HF and control groups (Figure 2).

### 3.3. ROC Curve Analysis of circRNAs with Differential Expression

The most significant differences in expression levels between the control and HF groups were observed in hsa_circ_0112085 (*p* = 0.0032), hsa_circ_0062960 (*p* = 0.0006), and hsa_circ_0053919 (*p* = 0.0074). To determine their diagnostic values in HF, we performed ROC curve analysis (Figure 3 and Table 3). The AUCs of hsa_circ_0062960, hsa_circ_0112085, and hsa_circ_0053919 for HF diagnosis were 0.838 ((0.740–0.937), *p* < 0.0001), 0.817 ((0.713–0.921), *p* < 0.0001), and 0.759 ((0.631–0.887), *p* = 0.001), respectively. The sensitivity and specificity values for the three circRNAs are shown in Table 3. Hsa_circ_0062960 showed the highest AUC and lowest *p*-values amongst the three circRNAs. Thus, it was chosen as the best-performing HF biomarker. The serum BNP level was strongly correlated with the expression of hsa_circ_0062960 (*R* = 0.649, *p* = 0.003; Figure 4).

### 3.4. GO and KEGG Pathway Analysis of CircRNA Genes

GO analysis was performed to investigate the biological processes of the host genes of all differentially expressed circRNAs as shown in the bar graph (Figure 5A). The most significantly enriched GO terms in the biological process were platelet activation (GO:0030168) and platelet aggregation (GO:0070527). The top 10 significantly enriched KEGG pathways of the circRNA genes are presented in Figure 5B. The results showed that the differentially expressed circRNA genes were strongly associated with platelet activation.

## 4. Discussion

In recent years, many circRNAs have been identified via high-throughput sequencing and bioinformatics analyses. CircRNAs, which are abundant and stably present in organisms, have become a focus of extensive research [21,22]. CircRNAs can modulate alternative splicing, transcription, and the expression of parental genes; one of the regulatory pathways involves microRNA (miRNA) sponges [23,24,25]. Differential circRNA profiles occur in patients with nonsmall cell lung cancer, primary biliary cholangitis, papillary thyroid carcinoma, and hypertension; certain circRNAs may serve as potential biomarkers in disease diagnosis [26,27,28]. A total of 15,318 circRNAs have been identified in the human heart, and highly expressed circRNA correspond to key cardiac genes, including titin, ryanodine receptor 2, and Duchenne muscular dystrophy [29]. CircRNAs are involved in heart diseases. For example, the heart-related circRNA was proven to act as an endogenous miR-223 sponge to modulate the expression of miR-223, through which it regulates cardiomyocyte hypertrophy [30]. Another circRNA that regulates cardiac function is *circ-Foxo3*, which is highly expressed in the cytoplasm of aged mice and patients [31]. Overexpression of circ-Foxo3 can induce fibroblasts senescence in vitro.

The failing heart exhibits huge variation in transcriptional scripts, which may predict disease progression and diagnosis. The identification and expression analysis of differential RNA molecules is a valid method for finding possible therapeutic targets and potential biomarkers [14]. Interestingly, increased circRNA expression is observed both in mouse and human failing heart tissues compared with nonfailing hearts [16]. Upregulated circRNAs in the myocardium may result in their increased expression in the circulation. This phenomenon was reported from miRNAs, another group of noncoding RNAs. For example, miR-423-5p is upregulated in human failing myocardium, and subsequently, in plasma [32]. Altered levels of miRNAs, such as miR-1 in the circulation of cardiac patients, can potentially be used as biomarkers of myocardial injury and HF [33]. The results of our current study were consistent with this notion from miRNAs, showing that increased levels of specific circRNAs in the circulation of patients with HF may be used as biomarkers for HF diagnosis. However, circRNAs are enriched and stable in whole blood, platelets, and plasma, thereby making circRNAs in human peripheral blood good candidates to improve the diagnostic and predictive values of HF. On the basis of these differentially circulated circRNAs, some precise biomarkers might be developed to improve risk stratification and predict the risk of HF-related death. Meanwhile, symptoms and physiology of cardiac disease have obvious differences between females and males, so the possible differences based upon gender should be considered in developing circRNA biomarkers. The new biomarkers must promote the development of precision medicine in HF.

This study investigated the differentially expressed circRNAs in the plasma of patients with HF and aimed to understand the role of circRNAs in HF pathogenesis. We identified 696 significantly differentially expressed circRNAs, including 477 upregulated and 219 downregulated circRNAs, in the HF group compared with the control group. We validated five circRNAs by qRT-PCR in a larger group. Amongst the candidate biomarkers, hsa_circ_0062960 exhibited the greatest significant difference in expression and the highest diagnostic value in patients with HF. Hsa_circ_0062960 demonstrated potential as a diagnostic biomarker for HF in clinical practice. Furthermore, hsa_circ_0062960 showed a strong correlation with the serum BNP level, which is a well-established biomarker for HF.

HF is a severe disease that increases the risk of venous thromboembolism, stroke, and sudden death amongst patients worldwide. HF is related to platelet activity. Gurbel et al. showed that platelet activity is heightened in 22% of outpatients with stable HF symptoms [34]. Other reports also identified that patients with HF exhibit increased whole blood aggregation and platelet-derived adhesion molecules [35,36]. Although increased platelet activity is observed in HF, the mechanism of platelet activation in HF, especially in relation to diagnosis and prognosis, requires further study. In the present study, the plasma circRNA expression profiles of HF and healthy controls were screened by circRNA microarrays and verified by qRT-PCR. Interestingly, GO and KEGG pathway analyses indicated that the most significantly enriched terms were related to platelet functions. Some significantly dysregulated circRNAs were linked to platelet activation (Table 4). For example, hsa_circ_0069197 is derived from the *WDR1* gene, which suppresses platelet activity. *WDR1* is related to the platelet-mediated pathogenesis of cardiovascular disease [37]. In addition, hsa_circ_0062960, a sensitive biomarker, comes from the *DEPDC5* gene, which is required for the development of heart, blood, and lymphatic vessels [38]. These results implied that circRNAs may be involved in the platelet activity of HF and potentially used to predict prognosis.

## 5. Conclusions

In summary, this study identified the expression profiles of circRNAs in HF based on plasma level. Five circRNAs were selected and verified in a case–control cohort. Circulating cell-free circRNAs may represent novel class biomarkers for HF.

## Figures and Tables

**Figure 1 biomolecules-10-00187-f001:**
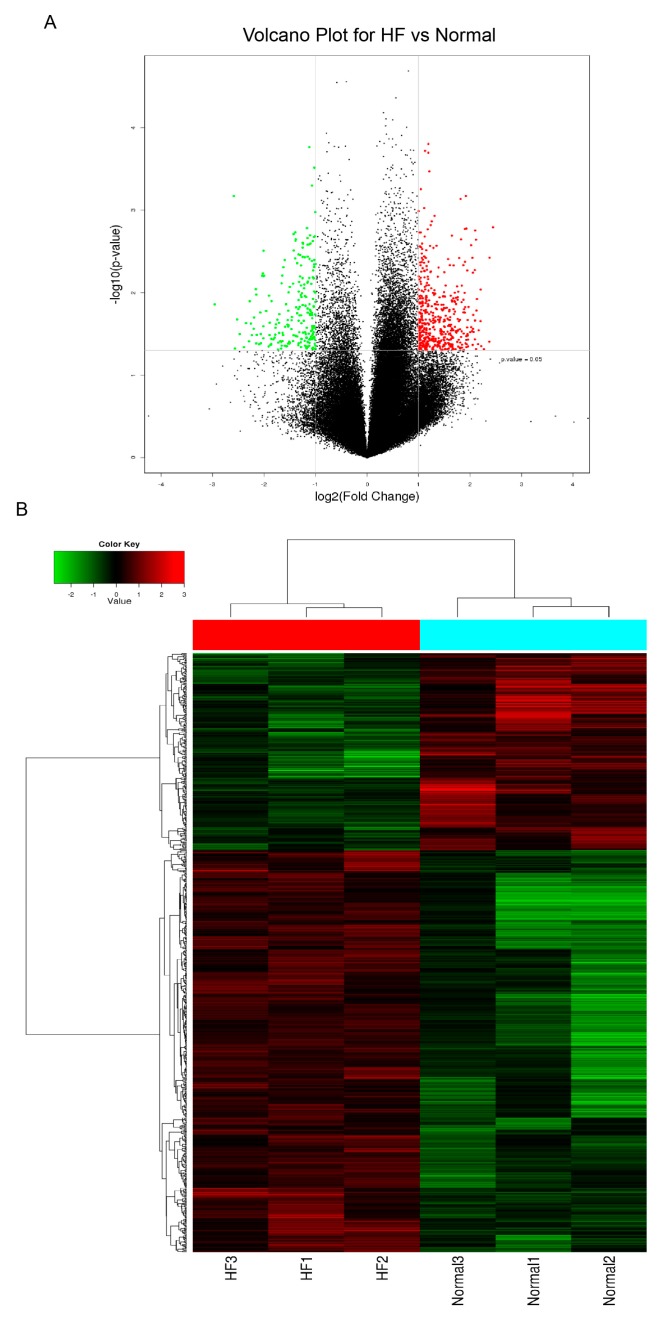
(**A**) Volcano plot of differentially expressed circular RNAs (circRNAs). Red plots represent overexpressed RNAs, and green plots represent downregulated RNAs with at least 2-fold change and corrected *p* < 0.05. (**B**) Heat map of the circRNA microarray profiles in the heart failure (HF) and control groups. The expression of circRNAs is hierarchically clustered on the *y*-axis, and blood samples are hierarchically clustered on the *x*-axis. Red indicates upregulated circRNAs, and green indicates downregulated circRNAs. Normal1–3 and HF1–3 are from control individuals and patients with HF, respectively.

**Figure 2 biomolecules-10-00187-f002:**
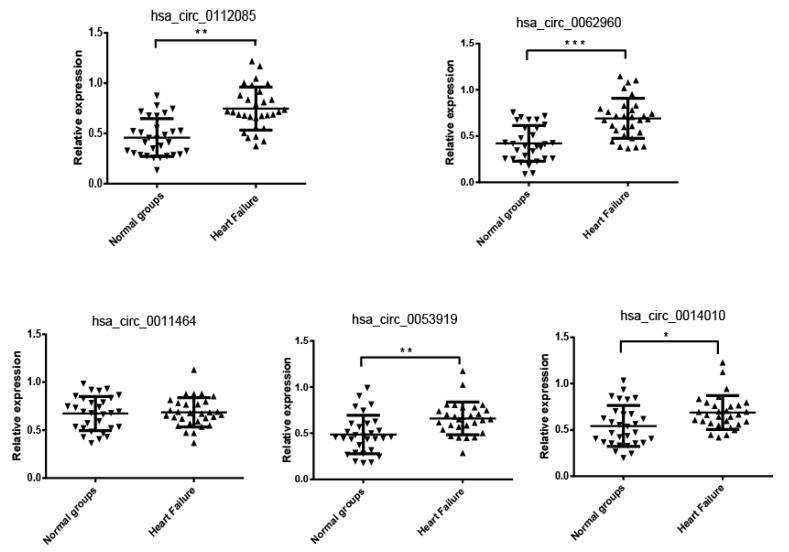
qRT-PCR validation for the expression level of hsa_circ_0112085, hsa_circ_0062960, hsa_circ_0011464, hsa_circ_0053919, and hsa_circ_0014010. The cohort included 30 patients with HF and 30 control individuals. The relative expression levels of circRNAs were normalized to levels of the control (U6 snRNA). Data are shown as the mean ± standard error of the mean (SEM). (* *p* < 0.05, ** *p* < 0.01 and *** *p* < 0.001).

**Figure 3 biomolecules-10-00187-f003:**
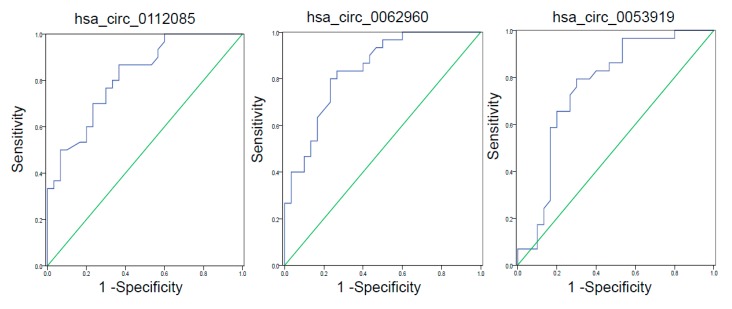
Receiver operating characteristic (ROC) curve analyses of circRNAs. The area under the curve (AUC) values are given on the graphs.

**Figure 4 biomolecules-10-00187-f004:**
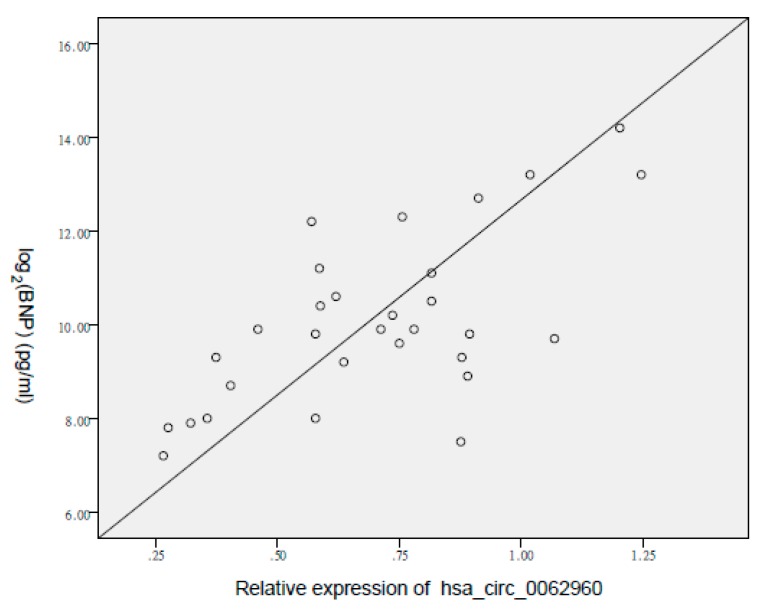
Correlation between the expression of hsa_circ_0062960 and serum brain natriuretic peptide (BNP) levels for patients with HF. BNP levels are displayed on a logarithmic scale. Pearson’s correlation is 0.649. The cohort included 30 patients with HF and 30 control individuals.

**Figure 5 biomolecules-10-00187-f005:**
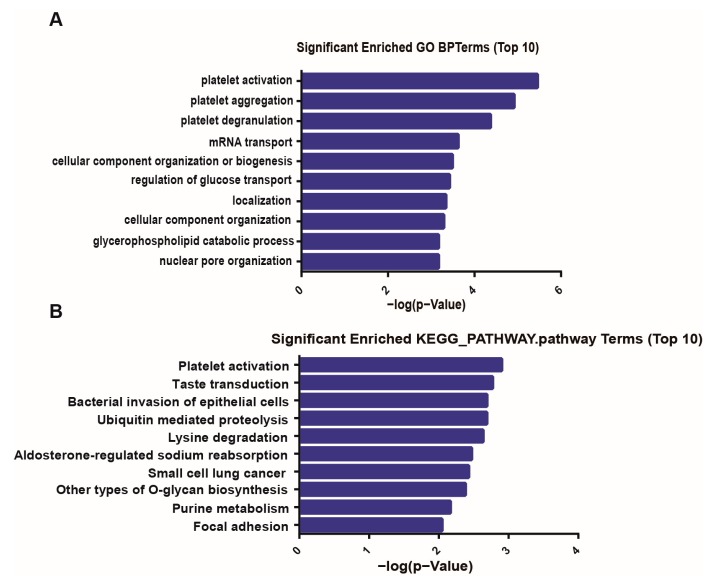
Functional annotations for the host genes of differentially expressed circRNAs. (**A**) The top 10 significantly enriched GO terms of biological processes of differentially expressed circRNAs in the HF and control groups. The bar plot presents the enrichment scores (−log10 (*p*-value)). (**B**) The top 10 significantly enriched pathways of differentially expressed circRNA genes. Their scores (−log10 (*p*-value)) are listed.

**Table 1 biomolecules-10-00187-t001:** The clinical features of heart failure (HF) patients and control groups.

Characteristics	Patients Group (*n* = 30)	Control Group (*n* = 30)	*p*-Value
Gender (male)	24 (80.0%)	22 (73.3%)	0.65
Age (year)	65 ± 10.2	62 ± 8.7	0.39
BMI	28.7 ± 5.2	26.3 ± 4.8	0.22
Diabetes	13 (43.3%)	2 (6.7%)	0.008
Hypertension	16 (53.3%)	4 (13.3%)	0.009
BNP (pg/mL)	1070 (145–17000)	12 (0–23.8)	<0.0001
Ejection fraction (%)	33 (25–39)	59 (52–67)	<0.001
NYHA grade		NA	
ClassI	1 (3.3%)		
ClassII	4 (13.3%)		
ClassIII	20 (66.7%)		
ClassIV	5 (16.7%)		

BMI: body mass index; BNP: B-type natriuretic peptide; NTHA: New York Heat Association; NA: Not Applicable.

**Table 2 biomolecules-10-00187-t002:** Selected circular RNAs (circRNAs) primers for quantitative reverse transcription polymerase chain reaction (qRT-PCR).

CircRNA ID	Location	Gene Symbol	qRT-CR primers	Product Size(bp)
hsa_circ_0112085	chr1:220928288-220936392	*MARC2*	F: 5’-TATTGTGGTGACCGGCTGTG-3′ R: 5′-TGTTTGAGGAAGGCTGCTTG-3′	192
hsa_circ_0062960	chr22:32242819-32242953	*DEPDC5*	F: 5′-GGGAAGAAGGGAACCTCAGC-3′ R: 5′-GGGGTGCAGTTGACTCCAGA-3′	129
hsa_circ_0011464	chr1:33557650-33560314	*ADC*	F:5′-CGGCTCGTGTTTGAAATGGG-3′ R:5′-GGCGCAGATGATCTTACTGG-3′	162
hsa_circ_0053919	chr2:33488360-33586582	*LTBP1*	F:5′-CTGAAGCTGGTGGTGAGAAC-3′ R:5′-ACTTGGCTGGCACTAGACGT-3′	103
hsa_circ_0014010	chr1:150312873-150318612	*PRPF3*	F:5′-GGATGTCAACGTGGTAGTAGTGG-3′ R:5′-GTGCTTCCCTCCTTGTTTGT-3′	196

MARC2—mitochondrial amidoxime reducing component 2; DEPDC5—DEP domain containing 5; ADC—acetoacetate decarboxylase; LTBP1—latent transforming growth factor beta binding protein 1; PRPF3—pre-mRNA processing factor 3.

**Table 3 biomolecules-10-00187-t003:** Validation of the circRNAs in a large group by qRT-PCR (HF = 30, Healthy = 30).

circRNA	AUC	95% CI	*p*-value	Sensitivity	Specificity
hsa_circ_0112085	0.817	0.713–0.921	<0.0001	0.867	0.533
hsa_circ_0062960	0.838	0.740–0.937	<0.0001	0.867	0.400
hsa_circ_0053919	0.759	0.631–0.887	0.001	0.862	0.467

CI: confidence interval

**Table 4 biomolecules-10-00187-t004:** Selection of eight significant circRNAs linked to platelet activation.

CircRNAID	Chromsome	CircStart	CircEnd	Strand	Gene	Ensembl ID	Reference
hsa_circ_0069197	chr4	10075962	10099515	-	*WDR1*	ENSG00000071127	[37]
hsa-circRNA14488-21	chr5	151054172	151055762	-	*SPARC*	ENSG00000113140	[39]
hsa_circ_0027463	chr12	69044179	69045831	+	*RAP1B*	ENSG00000127314	[40]
hsa_circ_0088697	chr9	130438922	130439032	+	*STXBP1*	ENSG00000136854	[41]
hsa-circRNA11337-4	chr17	8808093	8812491	-	*PIK3R5*	ENSG00000141506	[42]
hsa_circ_0001966	chr21	27347382	27372497	-	*APP*	ENSG00000142192	[43]
hsa_circ_0085166	chr8	101936362	101936526	-	*YWHAZ*	ENSG00000164924	[44]
hsa_circ_0091966	chrX	153587613	153590516	-	*FLNA*	ENSG00000196924	[45]

WDR1—WD repeat domain 1; SPARC—secreted protein acidic and rich in cysteine; RAP1B—RAS-related protein RAP1B; STXBP1—syntaxin binding protein 1; PIK3R5—phosphoinositide-3-kinase regulatory subunit 5; APP—amyloid beta precursor protein; YWHAZ—tyrosine 3-mono-oxygenase/tryptophan 5-mono-oxygenase activation protein zeta; FLNA—Filamin A.

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
