# Peer review of "Circular RNA Expression Profiles in Plasma from Patients with Heart Failure Related to Platelet Activity"

_biomolecules, 2020, doi:10.3390/biom10020187_

Round 1
Reviewer 1 Report
the authors have addressed all concerns
Reviewer 2 Report
The authors have properly addressed my concerns. I have no further comments.
This manuscript is a resubmission of an earlier submission. The following is a list of the peer review reports and author responses from that submission.
Round 1
Reviewer 1 Report
This is a well written paper with clear objectives and well thought out experimental procedures to address the issue. However, there are several points in the interpretation of the results that are not addressed and that need to be included in the Discussion.
- It is well established that symptoms and physiology of cardiac disease can be very different in females versus males. Since the majority of your subjects are males this could skew your results. At the very least mention of possible differences based upon gender should be addressed for future studies.
- There is a huge variation in the concentration of BNP in your HF population, does this correlate with the NYHA grade, and/or the presence of specific circRNA species?
- What would the advantage be of detecting specific circRNAs versus BNP concentrations as markers of HF?
- While the mean concentrations of selected cirRNAs are significantly different between the entire cohort of HF patients and controls there is also significant overlap in the high values and low values in both groups (Figure 2) which would indicate that if used for diagnostic purposes there would be a large number of false positives and false negatives and conclusions about the role of circRNAs in the pathogenesis of HF more difficult.
- In lines 229,230 it is stated that “hsa_circ_0062960 showed a strong correlation with the serum BNP”. Where is the data?
Minor points:
- Line 94 “differentially circRNA” is not proper English.
- Line 117. One normally uses standard deviations OR standard error, not both in data manipulations. Also, what method of reporting (SD or SE) was used for your different data sets was not indicated.
Reviewer 2 Report
It is an interesting paper, dealing with an appealing topic. It is a well performed study, with potential relevant clinical implications. Study figures are interesting. I offer the following comments.
How were healthy controls defined? Was an echocardiogram performed in them? The lack of a previous cardiovascular history does not necessarily exclude the presence of a silent myocardial dysfunction. Please, provide more details. Heart failure is a very heterogeneous population and two main entities have been distinguished: preserved and reduced (<40%) ejection fraction. By looking at Table 1, it seems that only reduced EF HF have been included and this should be specified in the methods.HF Can the authors specify the timing for their blood sampling? During HF stability or during admission for acute decompensated HF? As some patients were in NYHA class III/IV and upper BNP interquartile range include very high BNP values, this would suggest ongoing acute destabilization. The authors may better clarify these issues? Statistical analysis is robust. I would be interested to compare the AUC of the investigated biomarkers with that of BNP and whether they add on the diagnostic accuracy carried by BNP values? Net reclassification analysis may be of help, in order to provide the readers with new insights on the reclassification power carried by the investigated circRNAs. Any correlation between circRNAs expression and ejection fraction values? Any data on renal function and/or sodium levels? Any data on baseline therapy and whether this might have influenced the study results? Any data on the origin of EF reduction (ischemic/idiopathic)? In ischemic cardiomyopathy, aspirin is usually prescribed and this might have influenced the findings on platelet activity.
Reviewer 3 Report
Sun ET AL. evaluated the potential of circRNAs as biomarkers of HF. The topic is of interest. Nevertheless, the study has strong limitations that preclude its publication:
The use of GAPDH for normalization is unacceptable. This gene cannot be use for normalization of circulating RNA since it is not found in plasma. In case it is found is due to hemolysis … which is a confounding factor. The sample size is small. In the screening study, 3 HF patients and 3 control subjects were compared. In addition, the comparison diseased patient vs healthy control is not recommended for biomarker studies. The conclusions are overstated with this study design. It is fundamental to compare the potential of circRNAs as biomarkers with BNP and/or NT-proBNP. The AUC of both established biomarkers could give relevant information about the use of circRNAs in clinical practice. The use of bioinformatic analyses is useful. However, this should be use with caution. The results need to be validates experimentally. The authors should evaluate platelet activation to conclude that the levels of plasma circRNAs affect this function. Authors should be aware that the role of plasma RNA as endocrine mediators is still unclear.
Other comments:
Units are missing in Table 1. Missing information in abstract: sample size… Detailed information on RNA isolation and circRNAs quantification is missing. Additional experiments are needed to validate the primers: sequencing to detect back-splicing site… Statistical analysis: Software? Test to evaluate normalization? SE should not be used. Please use median (Q1-Q3) instead just median.Author Response
Please see the attachment.

Round 2
Reviewer 1 Report
Line 18: states microarrays done with 3 patients and 3 controls BUT your methods states microarrays performed with "pooled samples from 30 subjects" (lines 277-281). this must be corrected.
line 77: should read ".... in plasma of patients with HF"
line 88-89: should read "....the patients and control groups"
line 92 should read "...HF patients..."
Reviewer 3 Report
The authors still use GAPDH for normalization. This is an unappropiate approach. The conclusions are not supported by the results.
